# Urea-Assisted Synthesis of Mesoporous TiO_2_ Photocatalysts for the Efficient Removal of Clofibric Acid from Water

**DOI:** 10.3390/ma14206035

**Published:** 2021-10-13

**Authors:** Lidia Favier, Amalia Maria Sescu, Elaziouti Abdelkader, Laurence Oughebbi Berthou, Doina Lutic

**Affiliations:** 1Ecole Nationale Supérieure de Chimie de Rennes, CNRS, ISCR–UMR6226, Univ Rennes, F-35000 Rennes, France; laurence.oughebbi@ensc-rennes.fr; 2Faculty of Chemical Engineering and Environmental Protection, “Gheorghe Asachi” Technical University of Iasi, 700050 Iasi, Romania; sescu.amaliamaria@gmail.com; 3Laboratoire des Sciences, Technologie et Génie des Procédés L.S.T.G.P, Université des Sciences et de la Technologie d’Oran Mohammed Boudiaf (USTO M.B), Oran 31000, Algeria; abdelkader.elaziouti@univ-usto.dz; 4Faculty of Chemistry, “Alexandru Ioan Cuza” University of Iasi, 700506 Iasi, Romania

**Keywords:** mesoporous titania, CTAB, urea, isopropanol, ethanol, photocatalytic degradation of clofibric acid

## Abstract

Mesoporous TiO_2_ photocatalysts intended for the advanced removal of clofibric acid (CA) from water were synthesized by the sol-gel method in a medium containing cetyl-trimethyl-ammonium bromide (CTAB) and urea, using either ethanol or isopropanol to dilute the TiO_2_ precursor. The activation of the samples was undertaken at 550, 650 and 750 °C. The XRD revealed that the nature of the solvent resulted in significant differences in the anatase-to-rutile ratios obtained at different temperatures. The specific surface area values were situated between 9 and 43 m^2^·g^−1^ and the band gap values were similar for all the samples. The photocatalytic activity of the prepared samples was examined for the degradation of CA, an emergent water contaminant. The photocatalytic tests performed under UV-A irradiation revealed that the photo-reactivity of these materials depends on the calcination temperature. The best results were obtained for the samples calcined at 750 °C, which showed high yields of CA elimination, as well as almost complete mineralization (over 95%) after 180 min of reaction. Good results in terms of catalyst reusability in the reaction were found for the catalyst showing the highest photo-reactivity. Therefore, the samples can be considered good candidates for future water remediation applications.

## 1. Introduction

Titanium dioxide is currently considered the most popular photocatalyst due to its convenient band gap value, easy availability, reasonable price and low toxicity. The applications of TiO_2_ in photocatalysis cover a wide area of practical interests: environmental purification [1], advanced water treatments [2,3], air purification from volatile organic compounds [4], water splitting for hydrogen production [5,6] and fabrication of photocatalytic membranes [7]. A brief search performed using the keywords “TiO_2_ photocatalyst review” on Google Scholar revealed over 90,000 results, of which 17,600 were published after 2017 [8]. The great interest in this material is mainly due to the limitations and drawbacks of the existing TiO_2_-based materials, especially the widely available commercial product P-25. The recombination of the electron–hole pair on the surface of TiO_2_; the poor affinity of photocatalysts towards hydrophobic organic pollutants; the aggregation of the nanosized particles, which prevents the active centers from receiving the light irradiation or, on the contrary, the particles’ scattering in the reaction mass; and the difficult and deficient recovery from the treated water are just some of the reasons cited in literature [3] to support the interest in developing other alternatives for obtaining new photocatalytic TiO_2_-based materials.

Since the adsorption of the target compound on the surface of the photocatalyst is a mandatory step for the success of its oxidative degradation [9], the synthesis of mesoporous TiO_2_ has gained great attention in recent years. Mesoporous materials are valuable alternatives for the formulation of active heterogeneous catalysts, offering high specific area values and wide pores, which do not impose diffusion restrictions on the species to be adsorbed [10,11,12]. Therefore, after the discovery of siliceous mesoporous materials [13] in the 1990s, researchers’ attention was focused on obtaining non-silica mesoporous materials, and the objective of obtaining of mesoporous titania received special attention. Thus, a large palette of reaction pathways, including the sol-gel method [14,15], soft and hard template synthesis [15,16,17], the hydrothermal method [18] and evaporation-induced self-assembly (EISA) [19,20], were applied to obtain either pure titanium oxide or titanium oxides doped with various oxides, noble metals or carbon nanotubes. An excellent review by Gu and Schüth [21] brings together information about many of the systems, synthesis strategies and solutions used in obtaining non-silica mesoporous materials. Various template types, including amines, quaternary ammonium salts (CTAB), non-ionic surfactants (polyether block copolymers: P123, F127), esthers (Tween 80), alcohols, dodecyl-sulfate and numerous combinations thereof have been involved in obtaining mesoporous TiO_2_ with various degrees of crystallinity and specific structure types, in consonance with the detailed mode of structuration of the template and the detailed preparation procedure.

The soft templating method involves using molecules with structures and compositions that are able to establish high numbers of intermolecular bonds in order to generate ordered bulk units and then using these associations as casts to form a mineral or mixed mineral–organic network around them. The templating molecules group as lamellar structures, spheres or rods, depending on the solvent medium where they evolve and their concentration [22]. Due to the rich opportunities to establish hydrogen bonds, van der Waals forces and electrostatic intermolecular interactions between the external perimeters of the template units and the precursors of the targeted mesoporous solid, the network of the targeted solid product develops rapidly around the template assemblies.

Cetyl-trimethyl-ammonium bromide (CTAB) is a very versatile surfactant molecule for the structuration of various mineral walls around the micelles formed in aqueous systems. Cabrera et al. [23] reported successfully obtaining mesoporous titania from a titanium derivative containing triethanol-amine by deploying CTAB as a structure template in a strong alkaline medium (NaOH, pH around 10). The product had good thermal stability and thick walls compared to the similar silica structure MCM-41 (Mobil Composition of Matter No. 41). A neutral pH medium containing titanium butoxide and acetylacetone together with CTAB aqueous solution proved to be successful in obtaining mesoporous TiO_2_ with the conventional sol-gel technique and under microwave treatment [24]. When titanium oxysulfate-sulphuric acid hydrate was used as a titanium dioxide precursor, the conditions of an acid medium were established and mesoporous ordered structures were still obtained [25]. This template thus proved to be usable across a wide pH range.

When EISA is employed as the synthesis procedure, the hydrolysis condensation of the titanium precursor (organic salt of Ti) occurs slowly at mild temperatures (60 °C); a medium containing CTAB micelles is used and the reaction mixture is placed in opened Petri dishes. The structuration of titanium units occurs due to the good balance between the hydrolysis rate of the titanium salt and the condensation of the partially hydrolyzed species around the CTAB micelles [19,20], and it lasts 7 days. After this step, a mild hydrothermal treatment in ammonia solution is applied as a stabilization phase for 48 h [26,27].

In the current work, we report the synthesis of two samples of mesoporous TiO_2_, using CTAB as a structure templating agent, with a modified sol-gel technique. The template was left to settle the micelles’ formation in a mixture of water and alcohol (ethyl alcohol or isopropyl alcohol, the same species resulting from the hydrolysis of the titanium salt). The same alcohols were used in the two preparations for diluting the titanium tetra-isopropoxide (TTIP).

The success in obtaining an ordered product endowed with mesoporosity is conditioned by the delivery of the proper amount of small and reactive units of solid precursor around the template associations. These units were, in our work, the products of TTIP (Ti (OCH(CH_3_)_2_)_4_) hydrolysis, which occurred in four steps. The intermediate species were compounds of Ti units with one, two, three or four hydroxyl groups. In each hydrolysis step, isopropyl alcohol molecules were formed:Ti (OCH(CH_3_)_2_)_4_ + H_2_O → HO—Ti (OCH(CH_3_)_2_)_3_ + HO CH(CH_3_)_2_
HO—Ti (OCH(CH_3_)_2_)_3_ + H_2_O → (HO)_2_ Ti (OCH(CH_3_)_2_)_3_ + HO CH(CH_3_)_2_
(HO)_2_ Ti (OCH(CH_3_)_2_)_2_ + H_2_O → (HO)_3_ Ti (OCH(CH_3_)_2_) + HO CH(CH_3_)_2_
(HO)_3_ Ti (OCH(CH_3_)_2_) + H_2_O → (HO)_4_ Ti + HO CH(CH_3_)_2_

These Ti (OH)_x_ (OCH(CH_3_)_2_)_4−x_ species (x = 0–4) subsequently underwent condensation reactions with water release, leading to the formation of O–Ti–O units, the precursors of TiO_2_. When the hydrolysis and condensation reactions occur in a medium containing CTAB micelles, a late reaction occurs around them due to electrostatic interactions between the micelles and the Ti-containing species. This helps to generate mesoporous TiO_2_. The direct contact of pure TTIP with water generates very fast hydrolysis-condensation reactions and therefore the product formed is amorphous and virtually non-porous [28]. The dilution of the initial TTIP with an anhydrous alcohol (ethanol or isopropanol in our study) helps to slow the hydrolysis rate and increase the interaction with CTAB, promoting the formation of the mesoporous structure [29].

Urea was used in our work as a pH potentiator and implicitly as a species for controlling the hydrolysis-condensation ratio: at temperatures over 80 °C, two moles of carbon dioxide and one mole of ammonia result from the thermal decomposition of urea. We assumed that the ammonia resulting from this reaction could stabilize the mesoporous structure in a similar way as it does in EISA preparations. In comparison with the previous methods of Meynen et al. [26] and Beyers et al. [27], our approach involves the alternative of using a “green” molecule, urea. According to our literature search, urea was here used for the first time in a synthesis involving a soft template, delivering a time-saving procedure for obtaining mesoporous titania and performing the synthesis and structure stabilization steps in “one pot”.

The photocatalytic activity of the samples was evaluated through the degradation of a biorefractory emergent water pollutant, clofibric acid, under UV-A irradiation conditions. The obtained results were quite promising for potential practical applications from the point of view of the decomposition yields, mineralization extents and the reuse potential of the photocatalysts.

## 2. Materials and Methods

### 2.1. Chemicals

All the chemicals in the catalyst preparation were used without any preliminary purification. The titanium tetraisopropoxide Ti(iPrO)_4_ (TTIP) and cetyl-trimethyl-ammonium bromide (CTAB) were Sigma-Aldrich products. Ethyl alcohol (Et OH) and isopropyl alcohol (iPr OH) were purchased from Chemical Company (Iași, Romania). The clofibric acid used as model molecule for the investigation of catalyst activity was purchased from Acros Organics (Fair Lawn, NJ, USA). All these chemicals were used as received from the providers. Distilled water was used in all preparations and for the experiments carried out to investigate photocatalytic activity.

### 2.2. Catalyst Preparation

The samples were synthesized with the sol-gel and soft template method, using mixtures containing organic titanium salt (TTIP), alcohol (diluting agent) and water (for TTIP hydrolysis), as well as a surfactant (CTAB; template for the generation of a mesoporous structure) and urea (pH tuner and stabilizer of the TiO_2_ wall). Similar overall compositions were used for the two series of samples obtained in the presence of ethyl alcohol and isopropyl alcohol, which could be described by the below molar ratios between the reagents:

TTIP: CTAB: Alc OH: H_2_O: CO(NH_2_)_2_ = 1: 0.2: 20: 40: 5

Alc OH = Et OH for the E series and Alc = iPr OH for the P series of samples.

An example of the preparation protocol (E series) is presented below:(1)Mixture A: First, 9.1 g CTAB was dissolved for about 30 min in 55 mL H_2_O under magnetic stirring (500 rpm), then 73 mL EtOH was added and the agitation was continued during the preparation of mixture B;(2)Mixture B: For this mixture, 35.5 g TTIP was magnetically stirred with 72 mL EtOH for 30 min at 1000 rpm;(3)Mixture C: Mixture B was added in drops to mixture A under continuous magnetic stirring;(4)Mixture D: A total of 37.5 g urea was dissolved in 35 mL of water under magnetic agitation and slight heating;(5)Mixture D was added dropwise to mixture C, under continuous stirring.

The resulting gel-like mixture was heated under reflux in an oil bath (125 °C) and under agitation (1000 rpm) for 16 h.

After cooling, the mixture was separated by centrifugation over 10 min at 4000 rpm, washed with distilled water, centrifuged again and dried overnight at 60 °C. After crushing in an agate mortar, portions of the solid were calcined at 550 °C, 650 °C and 750 °C, for 6 h at a heating rate of 1 °C/min. The procedure is summarized in Figure 1.

The P series was prepared in a similar manner, using isopropyl alcohol instead of ethyl alcohol.

The following samples names were used for the series in this paper: E-550, E-650, E-750, P-550, P-650 and P-750. They indicate, respectively, the solvent used in the preparations (E—ethanol, P—isopropanol) and the value of the calcination temperature in °C.

### 2.3. Characterization of the Solids

The XRD patterns were traced on a Shimadzu D6000 device (Shimadzu Corporation, Tokyo, Japan; 2 theta range: 5–70°) under CuKα radiation (λ = 1.5406 Å). The SEM images were obtained on a JEOL JSM 7100 F EDS EBSD Oxford microscope (JEOL JSM-7100, Jeol Ltd., Tokyo, Japan). Besides the identification of the structure type and phase purity calculation, the particle sizes were calculated using the Debye–Scherrer relation:Dp = 0.9 λ/(FWHM·cosθ)(1)
where FWHM is the full width at half maximum for the maximum at angle θ. The investigation of the porous structure of the solids was performed with pure nitrogen adsorption measurements at 77 K using a Nova 2200 apparatus (Quantachrome Instruments, Boynton Beach, FL, USA. The dedicated software of the machine made it possible to measure the BET surface area and the free pore volume determination directly, while applying the Barrett-Joyner-Halenda (BJH) model made it possible to determine the pore size distribution of the samples. The UV–DR spectra of solid samples were traced on a Shimadzu 1700 device, preparing the samples with MgO as a blank and dilution solid material. The IR spectra were traced on a Jasco FT/IR-6100 spectrometer (Jasco, Waltham, MA, USA).

### 2.4. Photocatalytic Activity Evaluation

The photocatalytic activity of the as-prepared TiO_2_ samples was investigated through photocatalytic degradation of clofibric acid (C_10_H_11_ClO_3_, CAS 882-09-7), a molecule frequently reported for its recalcitrance to conventional wastewater treatment [30,31,32].

The photocatalytic tests were performed in batch mode in a cylindrical glass reactor with a total volume of 400 mL and a working volume of 250 mL. All experiments were carried out at ambient temperature, under continuous stirring (650 rpm), at the native solution pH (about 6.3) and under UV-A irradiation conditions. UV-A was provided with a 9 W lamp (Phillips, Poland; maximum emission peak at 365 nm) placed in the center of the reactor. For each experiment, an aqueous solution with the initial pollutant concentration of 10 mg·L^−1^ was used. Prior to the photocatalytic investigation, the considered catalyst (0.5 g·L^−1^) was added and the reaction medium was left in the dark at a stirring rate of 650 rpm at ambient temperature for a period of 30 min to achieve the adsorption equilibrium. This kind of investigation is necessary to assign the adsorption effect on the degradation efficiency of the target molecule during the photocatalytic reaction. Afterwards, the solution was irradiated for 180 min for the evaluation of the catalyst efficiency. The lamp was preheated (30 min) prior to starting the photocatalytic reaction in order to obtain its maximum intensity at the moment of the photocatalytic experiment’s startup.

Samples were taken from the reaction medium, filtered through a polypropylene syringe filter (0.2 µm, PALL Life Science, New York, NY, USA) to separate the catalyst and then analyzed with high performance liquid chromatography (HPLC) to determine the residual clofibric acid concentration. The pollutant quantification was performed with an HPLC using a WATERS^®^ instrument (600, Milford, MA, USA). The analyte separation was achieved with a BEH C18 column (250 mm × 4.6 mm, 5 μm). The analytic system used was fitted with a diode array detector (DAD, WATERS™ 996, Milford, MA, USA) and the detection of the target molecule was performed at 230 nm. A mixture of acetonitrile/ultrapure water (60:40 *v*/*v*; 0.1% formic acid) was used as the mobile phase. The injection volume and applied flow rate were 50 µL and 1 mL·min^−1^, respectively. The retention time for CA obtained in isocratic mode was 6.4 min. An external calibration curve (0–10 mg·L^−1^ range) was considered for the calculation of the pollutant concentration. Freshly prepared standard solutions were used for this purpose. The instrumental limit of detection for clofibric acid was estimated according to the procedure described by Kadmi et al. [33].

The total organic carbon (TOC) was measured using a Shimadzu TOC-5000-A system (Shimadzu Corporation, Kyoto, Japan) to estimate the mineralization level of the target molecule at the end of the photocatalytic reaction.

The abatement of the target molecule was calculated as:(2)$$Degradation\;\left(\%\right)=\frac{CA_{0}-CA_{t}}{CA_{0}}\cdot 100$$
where *CA*_0_ and *CA_t_* were, respectively, the pollutant concentrations at time zero and at time t of the reaction.

Also, the pollutant mineralization efficiency was calculated as follows:(3)$$Mineralization\;\left(\%\right)=\frac{\left[TOC_{0}\right]-\left[TOC_{t}\right]}{\left[TOC_{0}\right]}\cdot 100$$
where $\left[TOC_{0}\right]$ and $\left[TOC_{t}\right]$ were the concentrations of total organic carbon (mg·L^−1^) of the initial pollutant solution and the concentration measured after an irradiation time t.

All photocatalytic investigations were conducted in triplicate and the averages of the collected data were considered.

## 3. Results and Discussion

### 3.1. Structural and Textural Investigations

X-ray diffraction is an outstanding investigation method used to highlight the extent of the crystallinity of solid samples and calculate the phase purity, provided by a convenient database of patterns of already-known solids. The XRD patterns of the samples investigated in this work are shown in Figure 2. The XRD patterns indicated the presence of large majorities for the anatase phase and small amounts of rutile in all samples. The detailed assignments of the maxima are presented in Table 1.

According to the XRD, there were some differences between the two series of samples (E and P) in the crystallinity degree and the temperature value at which the formation of the rutile phase began.

The x_A_ anatase ratios in percent for the E and P series of samples were calculated by applying the relation deduced by Spurr and Myers [34], processing the XRD results for several series of anatase–rutile mixtures with different ratios between the two phases:x_A_ = 100 (1 + 1.26 I_R_/I_A_)^−1^(4)
where I_R_ and I_A_ are, respectively, the peaks areas for the most intense peaks related to the anatase ((101) plane, 25.4°) and to rutile ((110), 27.54°).

The crystallinity degrees of the samples were calculated by considering the best sample from the series as an internal standard [35] (P-750, the sample with the most intense maximum due to the (101) plane, was considered as 100% crystalline) using the relation:(5)$$\mathrm{Crystallinity}=\frac{1}{n\;}\sum_{1}^{n}\frac{\;I{\;}_{hkl\;sample}}{I_{\;hkl\;standard}}$$
where *I_hkl sample_* and *I_hkl standard_* were, respectively, the relative intensities corresponding to the most intense planes (101) relating to the anatase phase in the standard sample and in the current sample. The anatase ratios (%) in the samples and the values of the crystallinity degree are displayed in Table 2.

The ratio of the amorphous phase in the sample E-550 was 11.41% and was easily observed even prior to precise calculations due to the corresponding noisy baseline. The formation of some degree of rutile phase occurred even if calcination was undertaken at 550 °C, and the yield of the rutile phase clearly increased with the calcination temperature. For the P series of samples, the degree of crystallinity also increased with calcination temperature. At low calcination temperatures, the particle sizes were quite small, as indicated by the wider maximum relating to the (101) plane of the anatase phase (see also Appendix A). The calcination at 550 and 650 °C essentially only led to the formation of an anatase phase. Only small amounts of rutile phase (less than 5%) appeared from calcination at 750 °C, as confirmed by the low-intensity maxima relating to the (110) and (101) planes, giving signals at 27.5 and 36.2°. The use of the same alcohol (isopropyl alcohol) to dilute the TiO_2_ precursor as the one resulting from the hydrolysis of TTIP thus led to solids which had consistently higher thermal stability towards the formation of the rutile phase. These results were in line with those of Wetchakun and Phanichphant, who proved in their study that the formation of the rutile phase begins to show at temperature values between 700 and 800 °C for titania obtained with the sol-gel method in a manner similar to our work [36].

The particle size was calculated using the Debye–Scherrer equation (Table 3). The values were almost the same irrespective of the calcination temperature for the E samples, while significant differences appeared for P samples. Between 550 and 650 °C, the particle size almost doubled, and an important particle size increase was still noticed upon calcination at 750 °C. The XRD data revealed a strong influence from the alcohol used as solvent. The samples from the P series maintained high potential for deep structural transformation during the calcination and partly hindered the formation of the rutile phase.

A selection of the SEM images of the series of samples is provided in Figure 3 and Appendix A. All the samples consisted of irregular, polyhedral particles with similar sizes in the three space directions, tightly grouped together. At low calcination temperatures, some individual particles were rarely seen together in platelet-like agglomerations for the E samples.

During the exposure to higher temperatures, the grains became more regular and their arrangement looked like well-individualized stacks of rocks. An increase in the regularity of the piles of grains and a good homogenization of the aspects of the particles could be seen for both samples when calcined at 750 °C. In the case of the P series of samples, the particle sizes deduced from the XRD patterns using the Debye–Scherrer equation were evidently in poor agreement with those seen in the SEM images. For the samples calcined at 550 °C, it was somehow possible to see the particles that generated signals in the diffraction patterns as small grains here and there in the SEM images, but they grouped so closely together upon calcination at 750 °C that it was impossible to highlight their individual sizes.

The porous structure of the samples was investigated by using pure nitrogen adsorption at 77 K. The isotherms and the pore size distribution curves are displayed in Figure 4 and the textural data are given in Table 4.

The isotherms of the samples E-550 and P-550 could be appointed as type IV according to the IUPAC classification [37]. This behavior is specific to solids with porous structures that are able to adsorb in multilayers. The loops corresponding to samples E-550 and P-550 could be easily fitted as H2(b) types: the steep part of the desorption branch was attributed to the “ink bottle” pore shape [29]. The values of the specific surface areas determined by applying the BET equation were, respectively, 38.08 (E-550) and 43.63 (P-550) m^2^·g^−1^. The fitting of the BJH pore size distribution model made it possible to define the characteristic diameters for these samples, which were, respectively, 3.54 nm (E-550) and 3.88 nm (P-550).

The samples E-650 and P-650 demonstrated a pronounced porous structure compaction effect (collapse of the ordered mesopores). The appearance of the isotherms looked like that of type II, with a very narrow hysteresis of the H3 type. According to Thommes et al. [37], this isotherm shape is due to non-rigid aggregates of plate-like particles (also highlighted by the SEM images). There was a tremendous change in the desorption branch of the isotherm when the calcination temperature was switched to 650 °C. The specific surface area decreased dramatically, as well as the free pore volumes.

For the samples E-750 and P-750, the decrease in the adsorbed amount of nitrogen was even higher, and we could not record both the adsorption and desorption branches of the isotherms due to technical limitations of the apparatus; only the adsorption branch, making it possible to calculate the specific surface area with the BET model, was considered.

The micropore areas, deduced from the t-plot equation, suggested that the formation of some micropores between closely packed grains occurred during the calcination.

During the hydrolysis of TTIP, isopropanol was delivered in the reaction medium. Our results showed that the presence of a mixture between the in situ generated isopropanol and ethanol added as TTIP diluter in the system for the E-series samples seemed to be beneficial for the stability of the particle size, as seen from SEM images and XRD. In contrast, the use of isopropanol as diluting alcohol generated particles which showed a strong trend to increase in size at higher calcination temperatures. The stability of the mesoporous structure was also better in the case of the E series: upon calcination at 650 and 750 °C, the specific surface decreased by about three times, while in the P series the decrease amounted to almost five times. Isopropanol thus determined the evident collapse of the mesoporous structure and the intense increase of the particle size. The role of ethanol as a diluter of the reaction mixture also led to a more consolidated mesoporous system, maintaining similar particle sizes but losing the ordered porous architecture, while isopropanol was efficient in the initial definition of the mesoporous structure but ended in a sudden pore collapse and a strong reorganization of the particles, this being able to generate the important particle growth and strong tightening of the particles during the calcination.

The IR spectra of the samples are displayed in Figure 5. The broad band between 3600 and 3100 cm^−1^ and the band from 1637 cm^−1^ are due, respectively, to the-OH groups from adsorbed water on the surface of the solids and to the terminal hydroxyl groups bonded to the titania surface (bending modes of Ti–OH). It is worth noting that the intensity of the band associated with the adsorbed water was more intense for the sample calcined at 550 °C and decreased heavily when the calcination temperature became higher [38,39,40]. The absorption bands from 2359–2332, 2161 and 1979 cm^−1^ could be associated with some remains from the CTAB molecule after the calcination, respectively due to C–N bonding stretching. The same bands, slightly shifted, were also noticed by Wang et al. [39]. These bands did not change upon calcination, and the traces of impurities were bonded in a very stable manner on the titania support. This fact was also supported by the presence of small peaks around 1040 cm^−1^, which were attributed to Ti–O–C bonds and to the low intensity band from 1406 cm^−1^, due to both adsorbed water and N–H groups from the decomposition of CTAB [41].

The band gap values of the semiconducting samples of TiO_2_ were calculated after processing the UV–DR spectra, using the Kubelka–Munk (KM) model for data processing, by extrapolating the linear part of the curve and measuring the intercept with the abscissa. The KM curves are displayed in Figure 6.

The corresponding band gap values were measured by processing each transformed spectrum and are gathered in Table 5. The band gap values had close values between 3.12–3.24 eV and no correlation with the solvent used in the synthesis or the calcination temperature could be deduced. The results are in line with those reported by Zouzelka et al. [42] and those regularly reported for both rutile and anatase phases [43]. Furthermore, these data confirmed that all the as-prepared samples displayed band gaps situated between 3.1 and 3.24 eV, anticipating their photocatalytic activity in the UV range of the electromagnetic spectrum, as demonstrated by the photocatalytic evaluation experiments presented below.

### 3.2. Photocatalytic Activity of the Samples in the Degradation of Refractory Organic Pollutant

In this study, clofibric acid was used as target molecule to evaluate the photocatalytic performances of the two series of prepared photocatalysts (E and P series). The main parameters followed during our experimental investigation were the calcination temperature and the type of organic solvent employed for the catalyst synthesis.

Prior to the photocatalytic tests, blank experiments consisting of adsorption equilibrium investigations in the dark and photolytic experiments under irradiation conditions, in the absence of the catalyst, were conducted with the aim of evaluating the influence of these phenomena on the elimination of clofibric acid. The obtained experimental data revealed only a slight loss in the pollutant concentration during the blank tests, confirming the minimal contribution of these phenomena in the removal of this pollutant (data not shown). These results are in line with those reported in our previous studies for advanced clofibric acid removal from aqueous solution using other types of catalysts [44,45,46,47]. These studies confirm that the target molecule considered in our study exhibited a strong chemical stability, undergoing lower elimination by photolysis (less than 10%) and showing a very low adsorption capacity on the catalyst surface. Moreover, it was also found that the addition of the catalyst in the reaction media and the irradiation of the working solution play a positive role in the degradation of this molecule. All these aspects are considered in detail below.

In order to provide a complete investigation of the catalytic performance of the prepared samples, we firstly focused our attention on the evaluation of the photocatalytic activity of the E series. In this context, several photocatalytic tests were performed in the presence of the samples calcined at different temperatures. These assays were conducted for an initial pollutant concentration of 10 mg·L^−1^, a 0.5 g·L^−1^ catalyst load and under maximal irradiation conditions. Figure 7 depicts the obtained results for the runs assigned with this system as functions of the calcination temperature. Accordingly, the photocatalytic activity was enhanced by increasing the calcination temperature. Indeed, the system E-750 showed the best photodegradation efficiency, with 94% CA elimination within the first 30 min of irradiation, whereas only 74% and 82% CA abatement were achieved for the E-550 and E-650 systems, respectively. Figure 7 also provides the control results regarding CA degradation in aqueous solution with a commercial TiO_2_ catalyst named Aeroxide^®^ P25 (P25).

In addition, the E-750 sample achieved complete elimination of the target molecule after 60 min of reaction, which was very promising for the removal of this organic water pollutant. This result was also confirmed by the time-dependent HPLC chromatograms collected for CA degradation displayed in Figure 8. These results clearly demonstrate the high degradation efficiency of the target compound. Indeed, the recorded chromatogram and especially the measured peak area for clofibric acid strongly decreased after 30 and 60 min of reaction, respectively.

Today it is well-known that this bioactive metabolite of the lipid regulator drugs clofibrate, etofibrate and etofyllinclofibrate involves potential risk for ecosystems and human health and, unfortunately, because of its complex structure, it cannot be completely removed during the conventional biological treatment used for wastewater treatment processes [48,49,50].

A possible explanation for the superior photocatalytic activity observed with the E-750 system is that the rutile phase of the synthesized titania increased with the calcination temperature, which would be in line with the results reported by Wetchakun and Phanichphant [34]. Other studies have confirmed that biphasic titanium dioxide displays enhanced photoactivity [43]. Also, Yu and co-workers found that the heterojunction structure of anatase/rutile phases suppresses the recombination of photocarriers and, as a consequence, enhances photocatalytic activity [51]. From the point of view of photocatalysis principles, electron and hole recombination is considered to be detrimental to catalyst efficiency. For better semiconductor performance, these species must be effectively separated and charges should be quickly transferred to avoid their recombination [52]. Recent work dedicated to TiO_2_ photocatalysts reported that such phenomena occur when an excited electron drops back to the valence band through a passive radiative or non-radiative process and the charge interaction can take place at the surface or even in the bulk of the catalyst sample. Also, it was highlighted that this process is controlled by the mobility or trapping of e^−^/h^+^ charge carriers, the existence of an interfacial charge carrier transfer or the density of defects present in the semiconductor lattice [43,53,54].

Additional photocatalytic trials were conducted under the same operating conditions in the presence of the photocatalysts of the P series in order to assess whether the calcination temperature affected the photocatalytic activity of each system in relation to the degradation of the target molecule. It should be noted that the organic solvents used in this work for the synthesis of catalysts were selected based on their advantageous characteristics, such as their good efficiency as solvents for TTIP, helping to avoid the formation of amorphous titania when in contact with water in an undiluted state, or their low toxicity and easy availability. Regarding the samples from the P series, calcined at different temperatures, a similar trend was found as in the case of E series. The effect of increasing the calcination temperature on the photocatalytic performance of the investigated systems is clearly illustrated by the results depicted in Figure 7. The CA elimination was more pronounced for the experiments performed with the P-750 sample, showing a degradation rate of about 89% after 30 min of reaction and 95% at 60 min, respectively. In contrast, the P-550 and P-650 showed much lower photoactivity, with pollutant abatement in the range of 80–83% after a contact time of 60 min.

The most efficient sample prepared with ethyl alcohol was compared with the best sample from those obtained using isopropyl alcohol. As illustrated in Figure 7, both investigated samples showed quite similar photocatalytic activity for UV-A light-induced decomposition of CA. The use of the E-750 catalytic system led to a total pollutant elimination after 60 min of photocatalysis. Furthermore, the type of solvent employed during the titania synthesis seemed to play a minor role in relation to its catalytic activity in pollutant removal: only a small difference in catalyst performance (about 5%) was achieved for the P-750 sample. Accordingly, for this catalyst the photoreactivity was slightly reduced in the degradation of the target molecule compared to the samples of the E and P series (E-750 and P-750). However, the good photocatalytic activity of these systems may have resulted from the presence of phases relating to both anatase and rutile polymorphs. The ability of mixed titanium dioxide phases (anatase/rutile) to act as efficient photocatalysts in the oxidation of different emergent water pollutants from aqueous solutions has been described in numerous practical studies related to biphasic TiO_2_ [46,55,56].

The relatively higher photoactivity of mixed TiO_2_ phases compared to a single phase has been highlighted in several studies with regard to the decomposition of organic contaminants from water. In order to illustrate the difference in photocatalytic performance between mixed and single phases, Apopei et al. [57] isolated anatase and rutile from a commercial bilayer-type TiO_2_ catalyst (named Degusa P25), aiming for their dissolution in a solution of H_2_O_2_/NaOH, and evaluated their UV-A catalytic reactivity in the decomposition reaction of 4-clorophenol. According to their findings, the combined TiO_2_ phases showed notable photoactivity, while the separated crystalline phases (anatase or rutile) displayed lower catalytic efficiency in the oxidation of this molecule. The enhanced photocatalytic activity of the mixed phases was ascribed by the extended lifetime of the charge carriers.

It was surprising that, according to these results, no obvious correlations could be found between the photocatalytic performances and the band gap values or the specific surface area values of the catalysts. The lower photocatalytic efficiencies of the samples calcined at 550 °C could be easily connected to the presence of a pure anatase phase, but even these samples nevertheless exhibited quite good activity in relation to CA removal. The formation of the rutile was noted at 650 °C for the E sample and at 750 °C for the P sample. However, the photocatalytic performances were in the same range of values for both series, whether calcined at 650 or 750 °C. Since the ratio of the rutile phase was quite low in all the samples, the good catalytic results were rather unexpected. The photocatalytic process is promoted by the presence of heterojunctions between anatase and rutile, which are endowed with intermediate electronic levels and are able to host the electrons migrating back from the conduction level to the valence band [58]. Since rutile was scarcely present in our samples, as shown by the XRD, we assumed that it formed as small particles that were largely dispersed on anatase, ensuring the necessary surface conditions for electron–hole separation and for the analyte adsorption, even if the specific surface area was not very high. Similar behaviors were found by Zhou et al., who mentioned that the crystalline TiO_2_ had better activity than the less crystalline samples [59].

Besides the photodegradation efficiency of the prepared TiO_2_ frameworks, which depends on the calcination temperature, the reactivity of these catalysts in terms of CA mineralization is of utmost importance from the perspective of consideration for practical applications. Indeed, a high mineralization yield would be necessary for any possible future use of these materials for water purification purposes, because of the major issues in terms of the environmental and health risks posed by the presence of organic compounds and their intermediates (aromatic or non-aromatic), which are formed during the degradation of water pollutants. Therefore, TOC detection was also taken into consideration in our catalyst screening trials for both catalytic systems, in order to evaluate the degree of CA mineralization during the photocatalysis. Figure 9 summarizes the corresponding results for the mineralization efficiency in the E and P systems, respectively.

The most photoactive catalyst was E-750, which exhibited a substantial mineralization rate and achieved almost total elimination of the organic compounds (TOCs) after an irradiation time of 180 min. This result suggests that the CA and its organic reaction intermediates were almost totally transformed in carbon dioxide, whereas only a small amount (less than 3%) remained in the reaction media as organic compounds; this behavior is very promising for practical applications. For E-550 and E-650, the calculated mineralization yields were slightly lower (88% and 92%, respectively).

Furthermore, as shown in Figure 9, the data found for the photocatalysis with P frameworks indicated very similar behavior. In the case of this system, the hierarchy for mineralization activity was P-750 > P-650 > P-550. Also, in the same figure, it can be seen than the best samples from the E and P series displayed more positive roles in the mineralization of the target molecule than the plain TiO_2_ used as a control in our photocatalytic activity investigations.

Moreover, for the samples with lower mineralization efficiency, the results suggest that CA reaction intermediates still remained in the treated solutions after 180 min of reaction. It should be noted that the analysis of the recorded HPLC/DAD chromatograms of the samples collected from the reaction media at different time intervals showed that the photodegradation of the target molecule was certainly conducive to the formation of several reaction intermediates. The main ones detected were 4-chlorophenol and isobutyric acid, which were identified by comparing their characteristic retention times with those found during the analysis of a standard solution of each molecule (data not shown). However, their formation only occurred up to a certain reaction time and after that they remained in the treated solutions but a reduction in their peak area was observed because of their possible transformation or mineralization. In addition, the data presented in Figure 8 show that the CA was completely oxidized after 60 min of reaction for the sample (E-750 system) showing the best photoactivity. Furthermore, the high value achieved for TOC removal in the presence of the same sample after a reaction time of 180 min clearly confirms the almost total mineralization of the organic reaction intermediates.

The reusability potential of the optimal photocatalyst was also investigated as such an evaluation is essential to in order to check whether a catalyst is suitable for future practical applications. Hence, five successive photocatalytic tests were conducted for 60 min under UV-A irradiation conditions. After each cycle, the catalyst was recovered by filtration, washed, dried and reused in a new photocatalytic test. The obtained results for the cycling stability of E-750 are shown in Figure 10.

The E-750 catalyst displayed good potential for recyclability in terms of CA abatement during all considered cycles. Indeed, the calculated removal yields for CA were considered as consistent even after five successive runs. Only a slight loss (<6%) of the photodegradation efficiency was found after the fifth cycle. All these data clearly indicate the good capacity for reusability of the considered catalyst. The high degradation performance detected for the investigated sample over the five cyclic operations also confirmed the good physicochemical stability of this catalyst. Thus, the E-750 sample seems to be promising catalyst for future practical applications in water purification.

## 4. Conclusions

The results obtained in this work revealed that urea could successfully be used as a pH stabilizer, i.e., as an agent for regulating the hydrolysis-condensation ratio in obtaining mesoporous titanium dioxide from a titanium tetraisopropoxide precursor. The nature of the alcohol (ethyl alcohol, series E, and isopropyl alcohol, series P) used for the dilution of the precursor before mixing it with aqueous solutions played a significant role in relation to the structure, phase purity and thermal stability of the product: when isopropanol was used in the preparations, the formation of rutile phase was completely hindered if the calcination was made at temperatures below 650 °C.

The photocatalytic tests performed using clofibric acid as the target molecule to be decomposed revealed that the two series of photocatalysts, E and P, had similar results in terms of photocatalytic decomposition yields and mineralization extent. After 180 min of photocatalytic reaction, the mineralization yields were around or above 90% when using solids from the E series, while more than satisfactory values between 80–95% were obtained with the P series.

The results presented here clearly highlight that both mesoporous frameworks (E and P series) showed interesting photocatalytic activity in relation to the elimination of clofibric acid. Moreover, these data also highlight the significance of the calcination temperature in enhancing the photocatalytic performance. The investigated frameworks preserved very interesting photocatalytic activity and also deep mineralization efficiency.

## Figures and Tables

**Figure 1 materials-14-06035-f001:**
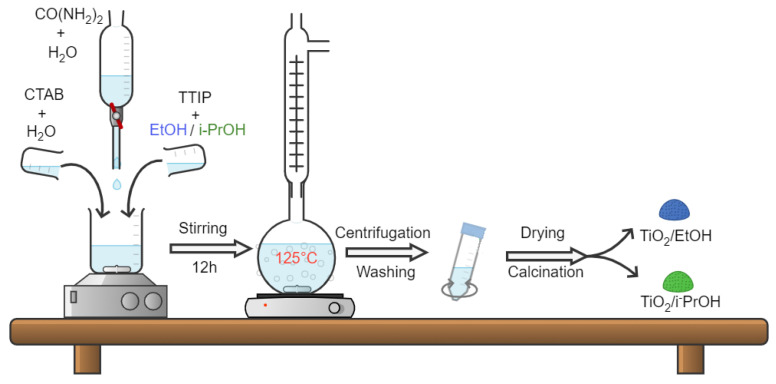
Schematic illustration of procedure used for the preparation of the mesoporous titania samples.

**Figure 2 materials-14-06035-f002:**
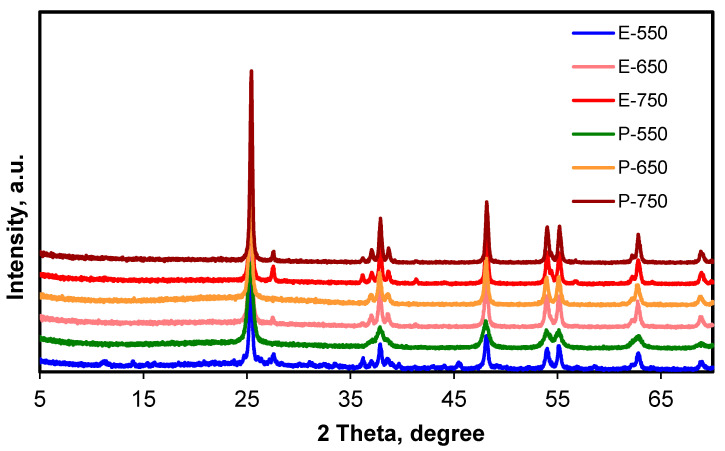
XRD patterns for the samples.

**Figure 3 materials-14-06035-f003:**
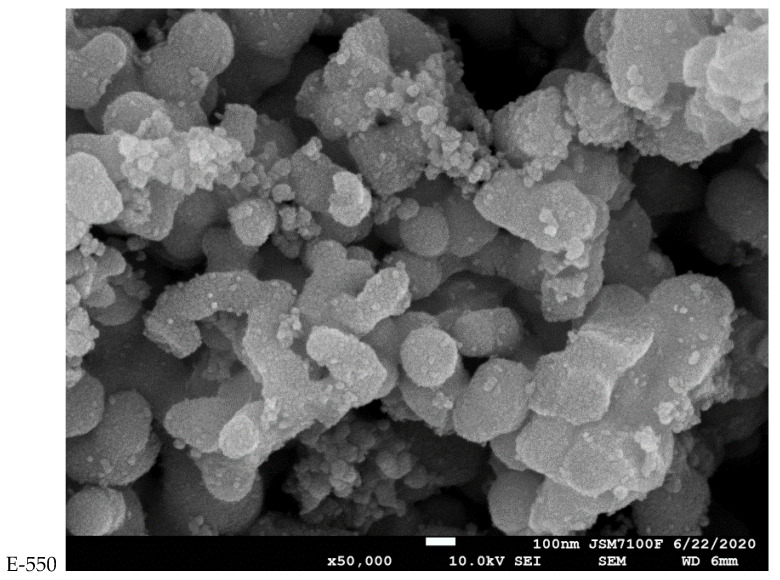
SEM images of the E and P samples calcined at 550, 650 and 750 °C.

**Figure 4 materials-14-06035-f004:**
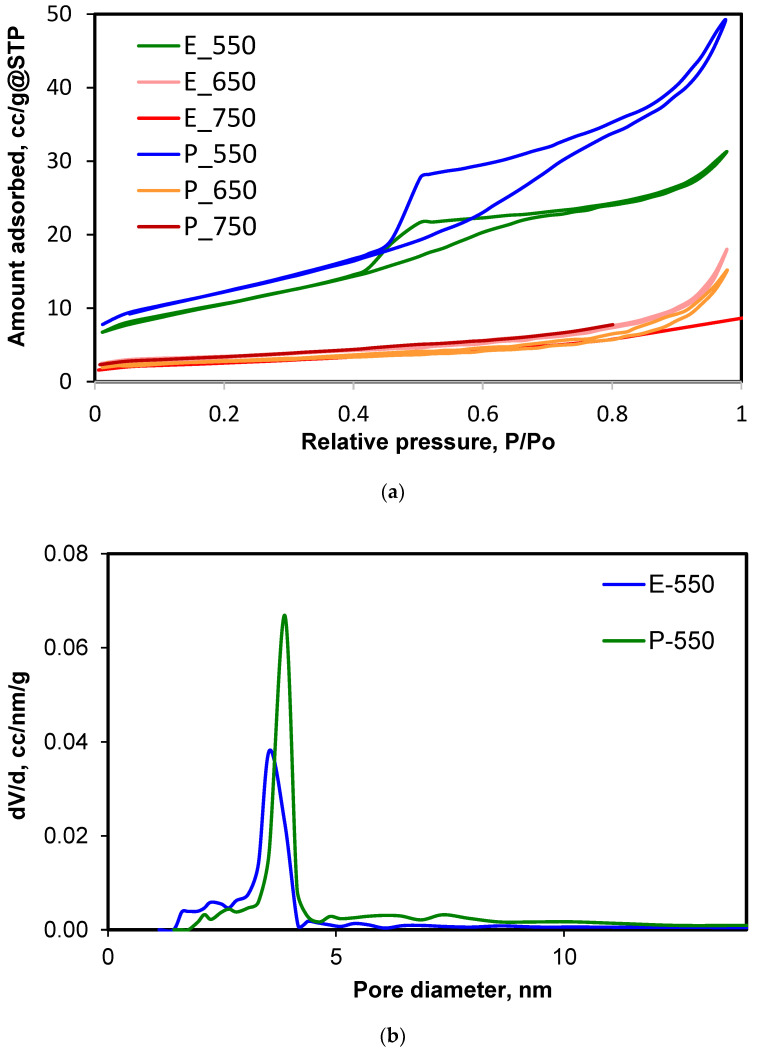
(**a**) Nitrogen adsorption isotherms at 77 K; (**b**) pore size distribution for samples E-550 and P-550.

**Figure 5 materials-14-06035-f005:**
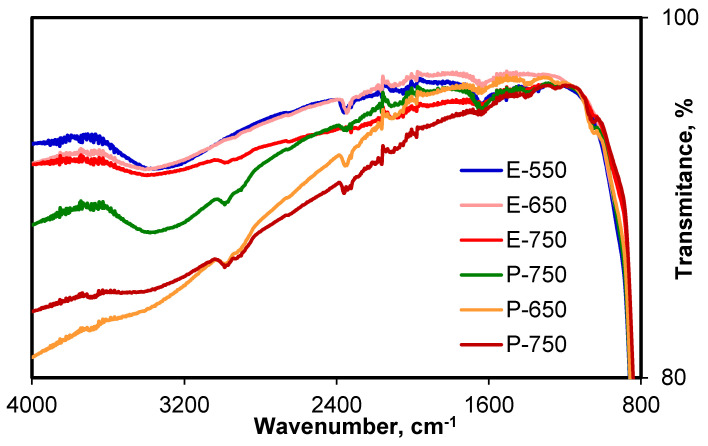
FT-IR spectra of the samples.

**Figure 6 materials-14-06035-f006:**
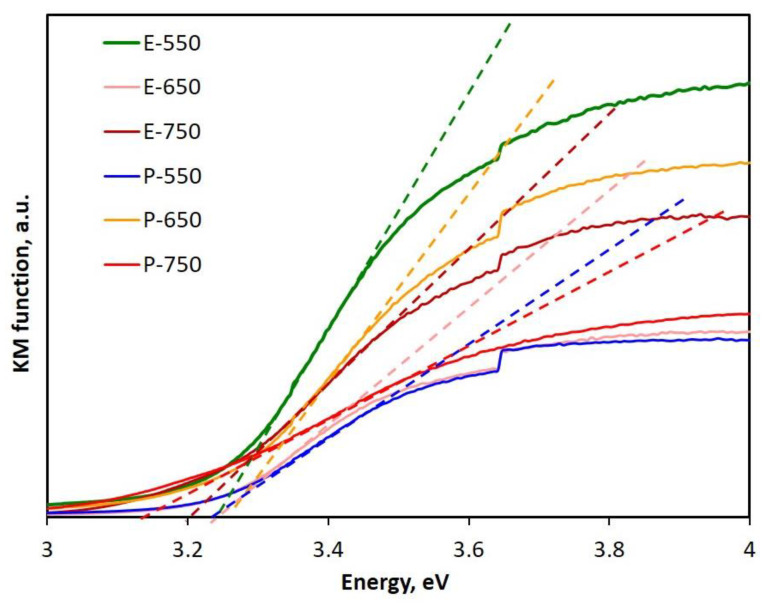
Deduction of band gap values from the Kubelka–Munk function (derived from the diffuse reflectance spectra of the samples).

**Figure 7 materials-14-06035-f007:**
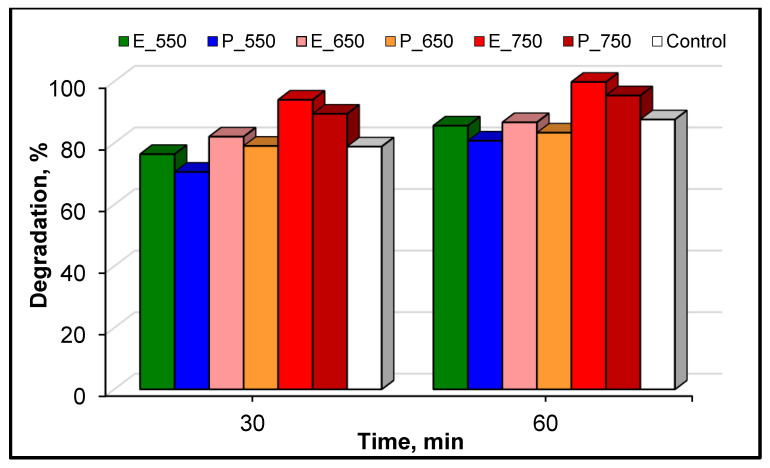
Photocatalytic activity of the TiO_2_ samples prepared with ethanol (E series), isopropyl alcohol (P series) and a control sample (P25) for the degradation of clofibric acid under UV-A irradiation. CA = 10 mg·L^−1^, catalyst = 0.5 g·L^−1^ and natural pH.

**Figure 8 materials-14-06035-f008:**
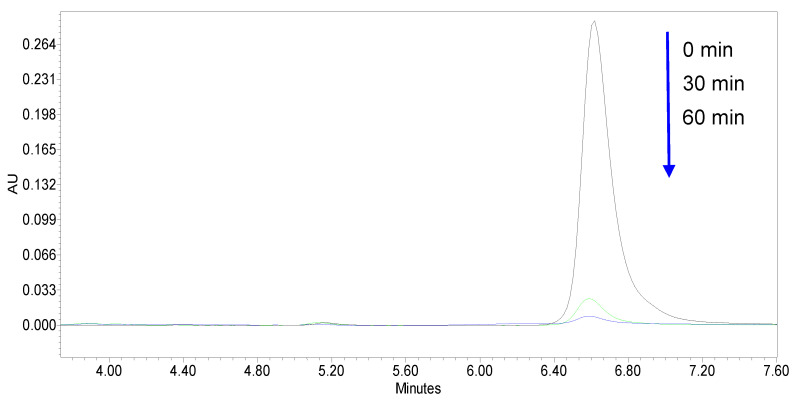
Time-dependent chromatogram of the measured HPLC peak for CA obtained for the samples collected with different reaction durations using the E-750 catalyst.

**Figure 9 materials-14-06035-f009:**
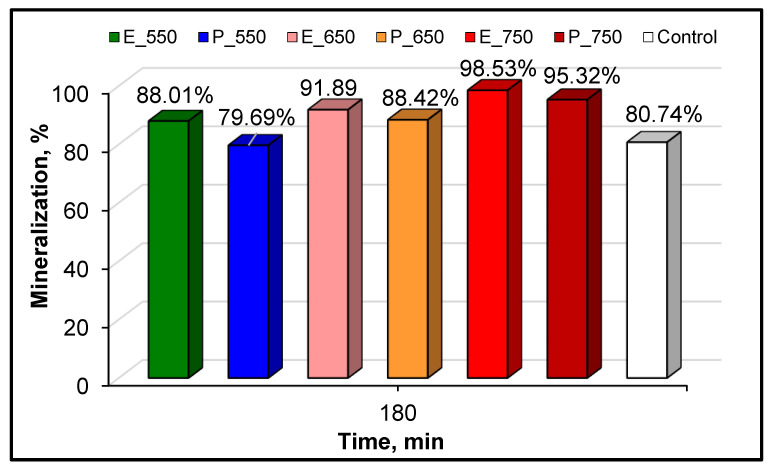
Mineralization degree of the as-prepared TiO_2_ samples calcined at different temperatures and the control after 180 min of UV-A irradiation. CA = 10 mg·L^−1^, catalyst = 0.5 g·L^−1^ and natural pH.

**Figure 10 materials-14-06035-f010:**
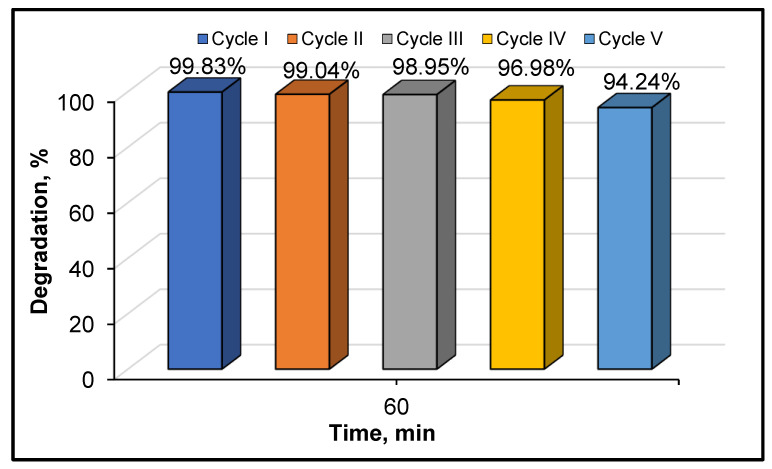
Photocatalytic degradation of CA by the E-750 system over five consecutive runs. CA = 10 mg·L^−1^, catalyst = 0.5 g·L^−1^ and natural pH.

**Table 1 materials-14-06035-t001:** Assignment of the signals from XRD.

Samples’ Maxima 2 Theta, Degree	Anatase Plane Index	Rutile Plane Index
25.40	(101)	
27.54		(110)
36.22		(101)
37.06	(102)	
37.88	(004)	
38.70	(112)	
41.40		(111)
48.14	(200)	
54.06	(105)	
54.48		(211)
55.20	(211)	
62.84	(204)	
68.84		(301)

**Table 2 materials-14-06035-t002:** Particle sizes calculated from the XRD pattern.

Sample	Anatase Ratio, %	Crystallinity, %
E-550	90.79	88.59
E-650	93.29	98.16
E-750	88.31	98.50
P-550	99.10	92.57
P-650	99.43	95.93
P-750	95.22	100.00

**Table 3 materials-14-06035-t003:** Particle sizes calculated from XRD patterns.

Sample	2 Theta, Degree	Size, nm
E-550	25.40	26.98
E-650	25.38	26.44
E-750	25.40	25.29
P-550	25.36	12.65
P-650	25.34	23.13
P-750	25.44	31.13

**Table 4 materials-14-06035-t004:** Textural properties of the samples calculated from the nitrogen adsorption isotherms *.

Samples	S_BET_, m^2^/g	Pore Volume, cm^3^/g	S_micropores_, m^2^/g
E-550	38.08	0.048	0
E-650	17.74	0.028	2.614
E-750	11.61	-	1.106
P-550	43.63	0.076	0
P-650	11.41	0.023	2.392
P-750	9.04	-	2.063

* Performed at 77 K, samples immersed in liquid nitrogen.

**Table 5 materials-14-06035-t005:** Band gap values of the samples calculated from UV–DR spectra.

Sample	Band Gap Value, eV
E-550	3.22
E-650	3.24
E-750	3.12
P-550	3.18
P-650	3.22
P-750	3.20

## Data Availability

Data is contained within the article.

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
