# Peer review of "Urea-Assisted Synthesis of Mesoporous TiO2 Photocatalysts for the Efficient Removal of Clofibric Acid from Water"

_materials, 2021, doi:10.3390/ma14206035_

Round 1

Reviewer 1 Report

This manuscript reports mesoporous TiO2 prepared by TTIP and urea.  The application target seems interesting, but for acceptance of this paper, this reviewer thinks it necessary to revise thoroughly by adding further experiments and/or sound discussion.  The detail comments are shown below.

(1) Authors examined physical structures of their materials by XRD, SEM, nitrogen adsorption, FT-IR and DRS.  However, in my opinion, authors revealed almost nothing about their materials and further experiments/analyses and/or sound discussion are needed to elucidate their materials. 

For examples, for XRD, authors are recommended, at least, calculating the ratios of anatase and rutile through Rietveld analysis for each material, and non-crystallinity (amount of amorphous titania), if possible, through using standard sample (e.g., Molecules, 19, 19573-19587 (2014))  Otherwise, the description "crystallinity extent", "phase purity" (L207), "there is some amorphous phase (L214)", "the yield of rutile phase increases clearly (L216-217)", " the degree of crystallinity is lower (L222-223)"cannot be mentioned.  Table 1 is not so important, instead, adding marks such as circles or squares indicating anatase and rutile above XRD pattern is more beneficial, which leads to easier understanding for readers, or inserting standard card as bars at the bottom of the XRD pattern is also adequate when showing XRD results.

For SEM images, how to display should be modified (scale bars).  Even though the authors claim their materials have (meso)porous structure, this reviewer cannot see it from the SEM images.  If authors would like to claim it, more clear images observed under higher resolution would be the evidence.  Current images cannot claim their differences in their morphology.

For nitrogen adsorption tests, this reviewer gets confused by the results and the current description.  First of all, the legend of Fig. 4 might be wrong.  Authors mentioned the hysteresis are categorized in type II.  If so, their materials don't have any pores or the size of pores should be more than 50 nm, that is not "mesoporous", but "macroporous".  The results of 750 ℃-calcined sample also are needed to be described in Fig. 4 by introducing second axis.  By calcination, it is easily expected that particle size would be changed, sintering occurs and pore size changes (or pore collapse happens), which all lead to the changes in specific surface area.  Therefore, in my opinion, it might be difficult to discuss the changes of particle size and mesoporous structure only from the results of nitrogen adsorption as discussed in L302-310. 

For FT-IR, this reviewer recommends adding the spectra of not-calcined samples as references.

For DRS, it is better to check legend again since the changes in calcination temperature seems random.  Also the caption of Fig. 6 should be checked. 

(2) Firstly, how to display the results of photocatalytic activity (Fig. 7-10) is rambling. (only 1 or 2 figures are enough to show that results.)  About the reason why 750 ℃-calcined sample showed higher activity (L385-400), it seems authors only find "convenient" papers that suit their results.  Generally speaking, anatase is better for organics decomposition due to higher conduction-band bottom, which allow one electron reduction of oxygen, which is a counter reaction of the oxidative organics decomposition.  (More importantly, authors did not evaluate the amount of rutile.)  Of course, adsorption of reaction compound (CA in this case) is very important for organics decomposition, which is strongly influenced on the specific surface area.  From the viewpoint of the diffusion of oxygen, smaller particle size might be preferable.  As pointed out frequently, photocatalytic activity of a material is not solely governed by its composition and there is no guarantee that the samples used in this study are representatives of TiO2, and they may one of the samples with the compositions.  In this sense, concluding the activity of materials with their composition is not preferable.  In this sense, the current manuscript did not discuss the reason of the activity properly based on the actual experimental results.  In addition, this is crucial thing, but the current results are uncertain whether their materials show higher photocatalytic activity or not, since authors used only materials prepared by them.  One of the methods to solve this point is to use commercial titania or so-called de-facto standard sample to be compared their photocatalytic activity with their samples.

(3) Since one of the main topic of this manuscript is using different medium: EtOH or iPrOH in preparation, the differences in (1) in-situ generated iPrOH and pre-added iPrOH, and (2) iPrOH and EtOH and how those alcohols affect the formation of the structure (mechanism) should be clarified and discussed in detail.  This reviewer cannot understand the description L311-316, which is not based on the actual experimental results.

(4) others: In addition, it is better to consider the essential figures again for all the results.

Current manuscript includes mix of "catalyst" and "photocatalyst".  Those terms are completely different.  Appropriate terminology is important.

This reviewer cannot find Figure A1 (L225) and Figure A2 (L244).

Sometimes authors explained only with "550 ℃" or "650 ℃" in text, which is ambiguous.  Reader cannot make sure whether TiO2/EtOH_550 or TiO2/iPrOH_550.  It is better to describe TiO2/EtOH_550 or TiO2/iPrOH_500 in the text.

About naming their sample, there is room for improvement (lengthy).  For example, TiO2/EtOH_550 change to E-550 and TiO2/iPrOH_550 to P-550.

English improvement is needed.

Reviewer 2 Report

The manuscript submitted by authors related to Mesoporous TiO2 synthesized in urea-containing medium for the efficient removal of persistent organic pollutants from water by photocatalysis is an interesting article and within the scope of Materials journal. But manuscript have some concerns.

  1. The title need to revise. Provide with the concise, attractive and informative title. 
  2. Abstract is not appropriate. Author have mention “ Photocatalysts based on mesoporous titanium dioxide were synthesized by the sol-gel method, in a medium containing cetyl-trimethyl-ammonium bromide (CTAB) and urea. Ethanol  and isopropanol were used in the two preparations to dilute the titanium isopropoxide used as Ti source’. This section is abstract section. Author need not to discuss the methods of synthesis under this section. Apart from that I will suggest the author to rewrite this abstract section again. Keeping following important point in knowledge. An abstract summarizes, usually in one paragraph of 300 words or less, the major aspects of the entire paper in a prescribed sequence that includes: 1) the overall purpose of the study and the research problem(s) you investigated; 2) the basic design of the study; 3) major findings or trends found as a result of your analysis; and, 4) a brief summary of your interpretations and conclusions.
  3. Under introduction section. Author have mention “The mesoporous materials are valuable alternatives for the formulation of active heterogeneous catalysts, offering high specific area values and wide pores, which do not pose diffusion restrictions to the species to be adsorbed.” Provide the refrence in support of this sentence.
  4. The author have synthesis mesoporous TiO2 by using CTAB as a structure templating agent, by a modified sol-gel technique. This is very common way of synthesizing this material. What is the novelty in this work. Need to be address.
  5. On page 3, section 2. Materials and Methods 1. Chemicals, made, grade and purity of chemicals are missing. Need to be added.
  6. Under section 2.2. Catalyst preparation, author have mention “The samples were synthesized by the sol-gel and soft template method’. What does author understand with soft template method? Need justification in details under discussion section.
  7. Author have used base catalyzed sol gel process or acid catalyzed sol gel process for the synthesis of photocatalyst.
  8. Provide the % Anatase phase and % Rutile phase in TiO2.
  9. Figure 3. SEM images of the samples TiO2/EtOH and TiO2/iPrOH calcined at 550, 650 and 750 oC, looks to be aggromelated. Will this agglomeration will have any impact to the efficiency of photocatalysts.
  10. Figure 5. FT-IR spectra of the samples looks to be very different. Can you re analysis the sample and denote all the major band in FTIR.
  11. Provide the detail for calculation of band gap?
  12. Provide the regeneration study of photocatalytic degradation of clofibric acid (CA)
  13. Will the pH will have any impact on the photocatalytic degradation of clofibric acid (CA)
  14. Provide the schematic mechanism for the photocatalytic degradation of clofibric acid (CA) and provide the details discussion under separate section “Mechanism of photocatalytic degradation of clofibric acid (CA)”.

Reviewer 3 Report

In this manuscript, the authors developed a mesoporous TiO2 photocatalysis to remove organic pollutants (clofibric acid) from water. Overall, the manuscript is well written, and the concept is interesting to the readers. the experiments seem to be carried out carefully and designed scientifically. I recommend this manuscript for publication with minor revision.

  • If possible, the author should include GC data to support the degradation of clofibric acid?.
  • Author should present the figure 7-figure 10 with positive control experiment, plain TiO2.

Round 2

Reviewer 2 Report

The authors have satisfactorily revised the manuscript.